# Assessment of Chemical Composition and Anti-Penicillium Activity of Vapours of Essential Oils from Abies Alba and Two Melaleuca Species in Food Model Systems

**DOI:** 10.3390/molecules27103101

**Published:** 2022-05-12

**Authors:** Veronika Valková, Hana Ďúranová, Nenad L. Vukovic, Milena Vukic, Maciej Kluz, Miroslava Kačániová

**Affiliations:** 1Institute of Horticulture, Faculty of Horticulture and Landscape Engineering, Slovak University of Agriculture, Tr. A. Hlinku 2, 94976 Nitra, Slovakia; veronika.valkova@uniag.sk; 2AgroBioTech Research Centre, Slovak University of Agriculture, Tr. A. Hlinku 2, 94976 Nitra, Slovakia; hana.duranova@uniag.sk; 3Department of Chemistry, Faculty of Science, University of Kragujevac, 34000 Kragujevac, Serbia; nvchem@yahoo.com (N.L.V.); milena.vukic@pmf.kg.ac.rs (M.V.); 4Department of Bioenergy, Food Technology and Microbiology, Institute of Food Technology and Nutrition, University of Rzeszow, 4 Zelwerowicza Str., 35-601 Rzeszow, Poland; kluczyk82@op.pl

**Keywords:** volatile compounds, in vitro antifungal activities, in situ efficacy, food model systems, disc diffusion method

## Abstract

The possibilities of the practical utilization of essential oils (EOs) from various plant species in the food industry have attracted the attention of the scientific community. Following our previous studies, the antifungal activities of three further commercial EOs, *Melaleuca armillaris* subsp. *armillaris* (rosalina; REO), *Melaleuca quinquenervia* (niaouli; NEO), and *Abies alba* (fir; FEO), were evaluated in the present research in respect to their chemical profiles, over four different concentrations, 62.5 μL/L, 125 μL/L, 250 μL/L, and 500 μL/L. The findings revealed that the major compounds of REO, NEO, and FEO were linalool (47.5%), 1,8-cineole (40.8%), and α-pinene (25.2%), respectively. In vitro antifungal determinations showed that the inhibition zones of a *Penicillium* spp. mycelial growth ranged from no inhibitory effectiveness (00.00 ± 00.00 mm) to 16.00 ± 1.00 mm, indicating a very strong antifungal activity which was detected against *P. citrinum* after the highest REO concentration exposure. Furthermore, the in situ antifungal efficacy of all EOs investigated was shown to be dose-dependent. In this sense, we have found that the highest concentration (500 µL/L) of REO, NEO, and FEO significantly reduced (*p* < 0.05) the growth of all *Penicillium* strains inoculated on the bread, carrot, and potato models. These results indicate that the investigated EOs may be promising innovative agents in order to extend the shelf life of different types of food products, such as bread, carrot and potato.

## 1. Introduction

In recent decades, there have been dramatic changes in demands for food quality and safety of consumers who are increasingly aware of the impact of food on their health [1]. Bakery products and various types of vegetables, as important constituents of the human diet, provide a substantial amount of essential nutrients [2,3]. However, these goods are mainly sensitive to microbial contamination reducing their shelf life [4]. In particular, the presence of microscopic filamentous fungi (including *Penicillium* spp.) can compromise human health due to the production of mycotoxins [5]. Moreover, it is well known that microorganisms have become increasingly resistant to common antifungals, as reported in numerous studies. Taking into account this trend, some ways of stabilization and preservation techniques, such as modified atmosphere, smart packaging, and bioactive or antifungal agent coatings, have been innovated [6]. For such purposes, plant essential oils (EOs) can serve as one of the possibilities of effective natural substance applications with antifungal activities [7]. In effect, their prediction of effective antimicrobials against yeasts and fungi have been reported in many studies [8,9,10].

Essential oils are mainly uncoloured fluids composed of volatile and aromatic substances which naturally occur in different parts of the plants (such as the stem, flowers, seeds, and peels) [11]. Owing to the complex chemical profile, their application appears to be a viable way in the prevention and elimination of antifungal food spoilage [12]. Indeed, the chemical composition of EOs is very complex, consisting of a mixture of more than 50 volatiles at very discrepant levels. Of these constituents, the most relevant for antifungal activities are terpenoid compounds and their derivatives, also designated as isoprenoids since the classification of terpenoids is based on the number of isoprene units [13]. There are several methods for the extraction of EOs, each exhibiting certain advantages and determining physicochemical and biological properties of the extracted oils [14]. The most widely used EO isolation techniques include traditional hydrodistillation, steam distillation extraction, organic solvent extraction, and microwave-assisted hydrodistillation [15].

Currently, more than 3000 EOs have been described, and only around 300 of them are of relevance for use in various industries. However, considering the enormous global diversity of medical plant species, as well as the industrial and commercial concern of EOs, this count is expected to increase radically [16].

Myrtaceae family, including more than 5500 species and approximately 150 genera, is considered the eighth largest flowering plant family with ecological and economic importance related to its production of EOs [17]. Among them, EOs obtained from *Melaleuca (**M.*) *armillaris* subsp. *armillaris* (rosalina) [18] and *M. quinquenervia* (niaouli) [19] have been shown to possess significant antifungal potential. Additionally, EOs of fir, *Abies* (*A.*) *alba*, belonging to the Pinaceae family has attracted an increasing interest for its distinctive and refreshing pine–forest fragrance. Having an easing and soothing effect on muscles, it is beneficial for the respiratory system [20]. Moreover, its strong antioxidant and antimicrobial activities also indicate its significant phytomedicine potencial [21].

In this context, as well as following our previous experiments to find new interesting naturally food antifungal agents, the aim of the present study is to characterize three commercially available EOs by evaluating their chemical composition and antifungal efficacies against selected microscopic filamentous fungi of genus *Penicillium* (*P. expansum*, *P. citrinum*, and *P. crustosum*). Finally, their use by the food industry to extend the shelf life of food products will be evaluated on food-based models (bread, carrot, and potato).

## 2. Results

### 2.1. Volatile Substances of EOs

All the EOs were analyzed by gas chromatography/mass spectrometry (GC/MS), and their detailed components are summarized in Table 1; in Table 2, the amounts of volatiles in percentage for each class of compounds are presented. From them, it is evident that the identified compounds represented 99.3%, 99.4%, and 99.7% of the oils from *M. armillaris* subsp. *armillaris*, *M. quinquenervia*, and *A. alba*, respectively. The most abundant compounds were shown to be linalool (47.5%), 1,8-cineole (16.9%), and α-terpineol (5.0%) in REO; 1,8-cineole (40.8%), α-terpineol (14.6%), and viridiflorol (12.0%) in NEO; α-pinene (25.2%), β-pinene (18.3%), and α-limonene (18.1%) in FEO.

### 2.2. In Vitro Antifungal Potential of EOs

In the current research, a disc diffusion method was applied to evaluate the antifungal activities of the selected EOs (REO, NEO, FEO) against *P. expansum*, *P. citrinum*, and *P. crustosum*. As shown in Table 3, the growth inhibition of the *Penicillium* strains depended on the type and concentration of the EO analyzed (*p* < 0.05); with increasing concentrations, the antifungal activities increased. A very strong antifungal effectiveness was observed for REO, where the highest concentration (500 µL/L) inhibited the growth of *P. citrinum* with an inhibition zone of 16.00 ± 1.00 mm. Moderate values for antifungal potential were detected for the highest concentration (500 µL/L) of REO and NEO against *P. expansum* (10.33 ± 0.58 mm) and *P. crustosum* (10.67 ± 0.58 mm), and *P. expansum* (11.00 ± 1.00 mm) and *P. citrinum* (10.00 ± 1.00 mm), respectively. The EOs exhibited weak zones of inhibition as follows: REO against *P. citrinum* (5.33 ± 1.53 mm in 125 µL/L; 9.33 ± 0.58 mm in 250 µL/L), and *P. crustosum* (5.33 ± 0.58 mm in 125 µL/L; 6.57 ± 0.58 mm in 250 µL/L); NEO against *P. expansum* (8.00 ± 1.00 mm in 250 µL/L), *P. citrinum* (5.33 ± 0.58 mm in 250 µL/L), and *P. crustosum* (7.33 ± 0.58 mm in 500 µL/L); FEO against *P. citrinum* (from 5.33 ± 0.58 mm in 125 µL/L to 7.67 ± 0.58 mm in 500 µL/L), *P. crustosum* (5.67 ± 1.15 mm in 250 µL/L and 8.33 ± 0.58 mm in 500 µL/L), and *P. expansum* (5.67 ± 1.15 mm in 500 µL/L). Remaining values for fungal growth inhibition indicated weak or very weak antifungal actions of the EOs.

### 2.3. Moisture Content and Water Activity of Food Models

Generally, the food quality depends on water activity (a_w_) and moisture content (MC) directly affecting the microbial growth connected with the shelf life of the foods. Both parameters were significantly different (*p* < 0.05) depending on the type of substrate analyzed (Table 4). In this line, values for MC ranged from 43.12 ± 0.35% (bread) to 86.83 ± 0.42% (carrot), and aw values varied from 0.942 ± 0.001 (bread) to 0.946 ± 0.002 (potato).

### 2.4. In Situ Antifungal Potential of EOs

In situ antifungal activity of all EOs investigated against filamentous fungi strains growing on three selected food models (bread, carrot, potato) was shown to be dose-dependent. In effect, we have found that the highest concentration (500 µL/L) of REO, NEO, and FEO significantly reduced (*p* < 0.05) the growth of *P. expansum*, *P. citrinum*, and *P. crustosum* inoculated on the bread model (Table 5). In *P. expansum*, the actions were additionally identified to be an EO type-dependent. In this sense, REO appeared to be the most effective (*p* < 0.05) in the fungus growth inhibition, and the EOs effectiveness decreased in the following order: REO > FEO > NEO. Against *P. citrinum*, the highest concentration of FEO exhibited the same antifungal activity as NEO and REO; however, both EOs (NEO, REO) had conversely stronger impacts (*p* < 0.05) on *P. crustosum* growth inhibition as compared to FEO.

The growth of *P. expansum*, *P. citrinum*, and *P. crustosum* on a carrot as a food model was completely inhibited by the highest concentration of REO (Table 6). Additionally, NEO was found to be able to inhibit the growth of *P. expansum* and *P. citrinum* but to a lesser extent (*p* < 0.05). On the other hand, it showed only a slight effectiveness against the growth of *P. crustosum*. FEO had the strongest antifungal action against *P. expansum,* which was similar to that of REO. By contrast, the inhibitory effect of FEO on the mycelial growth of *P. citrinum* and *P. crustosum* was lower than those of REO and NEO, and higher than that of NEO, respectively.

On a potato model (Table 7), the growth of *Penicillium* spp. was markedly inhibited by the highest concentrations of all EOs investigated (except for FEO in *P. crustosum*). Regarding this, the results showed the strongest antifungal activity of REO and NEO against the growth of *P. expansum*, *P. citrinum*, and *P. crustosum*. However, the inhibitory action of the highest concentration of FEO was species-dependent. Indeed, the EO displayed a very strong effectiveness against *P. expansum,* whilst its efficacy against the growth of *P. citrinum* and *P. crustosum* was only weak.

## 3. Discussion

Generally, it is well known that the antifungal activities of diverse EOs depend on their chemical profile and concentration of the individual constituents [22]. Regarding the *M. armillaris* subsp. *armillaris* EO, the literature reports the occurrence of three chemotypes of this species based on the proportions of 1,8-cineole, linalol, and methyleugenol [23]. According to Brophy and Doran [24], *M. armillaris* subsp. *armillaris* presents EO that is mainly monoterpenoid in character. The authors analyzed the EO samples obtained from two latitude areas. From their findings, it can be seen that the chemical composition of the samples was qualitatively very similar throughout the species range; however, quantitatively, the EO differed considerably in an apparent association with the latitude of collection. In line with our results, REO samples from the north of the species range contained linalool (56.2%) as the major component, and also significant amounts of 1,8-cineole (13.3%). On the other hand, REO from the south comprised 1,8-cineole (43.6%) as the principal component along with significant amounts of α-pinene (13.1%). Supporting our findings, linalool (41.6%), and 1,8-cineole (25.2%) as primary compounds of REO were also detected in the study by Zhao et al. [25].

The research of Ireland et al. [26], which examined the needle EO of *M. quinquenervia* over its geographical range in Australia and Papua New Guinea, has displayed a wide variation in chemical composition, and only two major EO chemotypes. Indeed, chemotype 1 was composed of E-nerolidol (74–95%) and linalool (14–30%), and chemotype 2 contained 1,8-cineole (10–75%), viridiflorol (13–66%), and α-terpineol (0.5–14%) in varying proportions and dominance order in the EO samples, which is in accordance with our findings. Similarly, Zhao et al. [25] found that 1,8-cineole (59.2%) and α-terpineol (9.6%) were primary compounds of NEO obtained from Australia.

FEO commonly contains various volatile substances, with α-pinene, β-pinene, camphene, limonene, and bornyl acetate being the most important [20,27,28]. In this context, the study by Zeneli et al. [29] showed the chemical variability of FEO from an Albania area, in which four samples of the genus *A. alba* were evaluated. Their results demonstrated that the samples were dominated on average by β-pinene (22.73%) and α-pinene (10.55%), which agrees with our study. Likewise, the studies by Roussis et al. [30] and Tsasi et al. [31] revealed β-pinene (19.8%) and α-pinene (10.9%), and β-pinene (26.9%), bornyl acetate (12.5%) and α-pinene (10.1%) to be the main components in FEOs obtained from the South Balkans and Greece areas, respectively.

Many studies indicated antifungal efficacy of different EOs types against a wide range of fungal strains such as *Penicillium* spp. (*P. polonicum*, *P. expansum*, *P. citrinum*, *P. crustosum*, *P. funiculosum*, *P. brevicompactum*, *P. glabrum*, *P. oxalicum*, *P. chrysogenum*) [32,33]. Very strong antifungal action of the highest REO concentration against *P. citrinum* growth observed in our study is reinforced by the work of Farag et al. [18], whose REO (in the concentration of 5 µL on the filter paper discs) exhibited the highest inhibitory effects (inhibition zone 18.3 mm) on the growth of *Aspergillus (A.) niger* among the EOs obtained from other *Melaleuca* species. We hypothesize that the very high antifungal activity of our REO may be due to the high proportion of linalool (47.5%). Indeed, results of the study by Dias et al. [34] suggest the antifungal efficacy of linalool itself against *Candida* (*C.*) *albicans*, *C. tropicalis*, and *C. krusei*. However, values for the minimal inhibitory concentration (MIC) differed among the strains tested. In this line, linalool displayed the lowest inhibitory concentration against *C. tropicalis* (500 μg/mL) followed by *C. albicans* (1000 μg/mL) and *C. krusei* (2000 μg/mL), and the growth of *C. tropicalis* was completely inhibited by linalool. The underlying mechanism of the linalool action is associated with its ability to damage the cell wall of microorganisms accompained by a reduction in membrane potential, leakage of alkaline phosphatase and the release of macromolecules (including DNA, RNA, and protein), how it was found in *Pseudomonas fluorescens* [35] and *Shewanella putrefaciens* [36]. Additionally, bacterial metabolic and oxidative respiratory perturbations interfering in cellular functions and even causing cell death have been demonstrated in these studies. The very strong antifungal activity of REO can also be attributed to the high proportion of 1,8-cineole (16.9%) in its composition, which is also known for its fungicidal properties. This fact was also confirmed by Vilela et al. [37] who recorded a partial inhibitory action (5.5%; independent of organism) of 1.35 µL of 1,8-cineole (isolated from *Eucalyptus globulus* leaves) on the mycelial growth of *A. flavus* and *A. parasiticus*, thereby recognizing the component to be one of those responsible for *E. globulus* antifungal activity. In this view, it can be concluded that the presence of this substance in our NEO (40.8%) may greatly contribute to its antifungal efficacy. In this regard, Tančínová et al. [38] analyzed three EOs from the genus *Melaleuca* (tea tree, cajeput, and niaouli). They have found that *M. alternifolia* Cheel (tea tree) EO showed a strong inhibitory effect (from 84.8% to 100% inhibition) on the strains of *P. commune*. A weaker inhibitory action was reported for *M. leucadendra* (cajeput) EO and the weakest one for NEO. As in our study, plant species- and pathogen species-dependent variation in in vitro antifungal potential of 11 Myrtaceae EOs has also been reported by Lee et al. [19]. Regarding 1,8-cineole, it has been found that *Staphylococcus aureus* treated with the chemical substance exhibited prominent outer membrane disintegration with a concentrated/reduced/agglomerated nucleoplasm [39]. Additionally, the study performed by Yu et al. [40] showed a destructive effect of 1,8-cineole on organelles along with the appearance of many unidentifiable vesicular structures in *Botrytis cinerea*. Using transmission electron microscopy, the authors found that 1,8-cineole may penetrate the cell membrane and damage cellular organelles without causing lesions on the membrane. Moreover, Nikolova et al. [41] observed very weak inhibitory activity of acetone extract from *A. alba* needles against the growth of *Alternaria alternata,* corresponding with our findings dealing with weak antifungal effectiveness of FEO against the selected *Penicillium* strains investigated. A remarkable selective antimycotic effect of FEO against a clinical strain of *C*. *albicans* has also been demonstrated in the study of Salamon et al. [42]. Although the underlying mechanism of antifungal action of the EOs has not been yet fully elucidated, we propose that in our EOs, it may be linked to the inhibition of microscopic filamentous fungi respiration and disruption of the permeability barriers of their cell membrane structures [43]. On the basis of the promising data from in vitro antifungal activity, the samples of our EOs were also applied to determine their vapour-phase inhibitory effects on the *Penicillium* spp. growth inoculated on the selected food models.

Moisture content and aw are the pivotal parameters in predicting the quality and stability of food products [44]. Moreover, they are very valuable indicators because their values commonly correlate well with the potential for fungi growth and metabolic activity [45]. Concretely, MC defines the amount of water in a product, thus, providing information about its yield, quantity, and texture; however, it does not provide credible data regarding microbial safety. On the other hand, aw expresses the volume of water that is available not only for reaction with other molecules but also for food spoilage processes including enzymatic browning and microbial growth. Taking into account these aspects, aw is an indicator of food stability with respect to microbial growth, biochemical reaction rates, and physical properties [46]. Generally, it is known that the growth of microorganisms occurs in product samples with aw values higher than 0.60 [47], which is consistent with our detected data. In line with these findings, the approximate value for aw in white bread is within the range of 0.94 to 0.97 [48], indicating its susceptibility to microbial spoilage with the main effect coming from the growth of various molds. With respect to our result obtained from the bread analysis, similar values for aw (0.944; 0.948) were observed in two bread samples from registered baking industries in the research by Ayub et al. [49]. Regarding the MC, bread is a food product with intermediate moisture [50], typically ranging from 35–42% [51], which also corresponds with our results. Nonetheless, most of the fresh foods are perishable because of their high values for MC [52], as it was also shown in our vegetable samples (>80%). Similarly to our results, Sipahioglu and Barringer [53] estimated the values for MC and aw in carrot and potato to be 89.97% and 0.996, and 75.19% and 0.990, respectively. All the mentioned findings indicate the suitability of chosen food models (bread, carrot, potato) for in situ antifungal analysis of the EOs investigated.

After testing the ability of the food spoilage fungi (*P. citrinum*, *P. crustosum*, and *P. expansum*) to grow on the selected food models, the antifungal efficacy of REO, NEO and FEO was evaluated. Generally, food products require protection against fungi deratoration during their storage [54]. In bakery products, the microscopic filamentous microorganisms (mainly *Penicillium* spp. and *Aspergillus* spp.) are the most common species causing their spoilage [55,56]. These fungi are able to grow in the bread surface and form a greenish-blue layer [57]. Besides the degraded quality including external appearance, the fungi are responsible for an unpleasant taste and aftertaste formation, and the production of mycotoxins and allergenic compounds. Importantly, these substances may be created even before the growth is visible [55]. Bakery goods usually have a short shelf life (only a few days at room temperature) due to their high aw, as it was confirmed in the previous part of this research. Spoilage of fruits and vegetables caused by fungi during storage is also the major concern affecting their quality and shortening their shelf life [54]. Therefore, the application of EOs in packaging may be the principal choice for satisfying and ecologically demanding ways to extend the shelf life of products without using synthetic preservatives.

The antimicrobial effect of EOs against various types of food spoilage microorganisms, after being applied by direct contact, has been demonstrated on a large scale. However, the vapour phase and volatile components present in EOs have not been thoroughly investigated [58]. The bioactivity of EOs in the vapour phase seems to be an interesting alternative that makes them potentially useful as antimicrobial agents for the preservation of stored fresh products. In effect, promising results in this field of research have been obtained, especially for bacteria and fungi [59,60,61]. Additionally, some studies have reported that vapour generated by EOs has a greater antimicrobial effect as compared to their liquid form applied by direct contact [62,63]. Moreover, Nadjib et al. [64] found that lipophilic molecules in the aqueous phase associate to form micelles restraining the attachment of EOs to microorganisms, whereas the vapour phase allows for free attachment.

The results of our in situ analysis revealed the antifungal effects of all three tested EOs (REO, NEO, and FEO) against selected *Penicillium* spp.; however, their effectiveness varied depending on diverse factors including the type of EOs, their concentrations, and also the fungi and food model used. Although *Penicillium* strains are known due to their high resistance [65], their mycelial growth on all three food models was suppressed by all EOs used with the strongest effectiveness in their highest concentration (500 µL/L). We suppose that the major compounds found in the EOs concept can primarily contribute to the antifungal effects. However, it is necessary to keep in mind that the overall EO effect is not attributed to just one or only some of its components [66], but it is a result of the synergistic action of all its constituents [67]. The antifungal efficacy of linalool (present in our REO in 57.5%) was also confirmed in the study performed by Xu et al. [68] who exogenously applied linalool (in the concentration of 20.95 μM and 2095 μM) on strawberry fruits infected with *Botrytis cinerea*. The antimicrobial potential of linalool has also been demonstrated in the research of Chang et al. [69] who evaluated the effect of active polyethylene (PE) film containing linalool active components (0–2%) on the microbial shelf life of mozzarella cheese. They found that PE films with a higher linalool content significantly suppressed fungal growth throughout the storage period (30 days). Moreover, the results obtained by Soković et al. [70] suggest that 1,8-cineole and α-pinene (primary constituents in our NEO and FEO compositions, respectively) are very strong antifungal agents with MICs of 2.00 μL/L (*Trichoderma harzianum*) and 7.00 μL/L (*Verticillium fungicola*), and 3.00 μL/L (*Trichoderma harzianum*) and 8.00 μL/L (*Verticillium fungicola*), respectively.

From the findings of all our analyses it can be concluded that REO, NEO, and FEO may be promising constituents with a potential use for extending the shelf life of bread and vegetables in the commercial scale of the food industry.

## 4. Materials and Methods

### 4.1. Tested EOs

Rosalina EO (REO; *M. armillaris* subsp. *armillaris*), niaouli EO (NEO; *M. quinquenervia*), and fir EO (FEO; *A. alba*) were extracted by steam distillation of fresh needles and leaves. These EOs were obtained by a commercial producer Hanus Ltd. (Nitra, Slovakia), and were preserved at 4 °C in the laboratory refrigerator until their next use. Importantly, the research complements our knowledge gained in previous similar experiments [33,71]. In this regard, it provides a comprehensive overview of the biological effects of different types of commercial EOs obtained from the same company.

### 4.2. Determination of EOs Volatile Constituents

The volatile compounds of the EOs were determined using gas chromatography with mass spectrometry (GC-MS), as it was described by Valková et al. [33]. In brief, the analysis was carried out by Agilent Technology 6890N (Agilent Technologies, Santa Clara, CA, USA) coupled to quadrupole mass spectrometer 5975B (Agilent Technologies, Santa Clara, CA, USA). Separation of compounds was carried out using HP-5MS capillary column (30 m × 0.25 mm × 0.25 m). The temperature program was as follows: 60 °C to 150 °C (increasing rate 3 °C/min) and 150 °C to 280 °C (increasing rate 5 °C/min), using helium 5.0 as the carrier gas with a flow rate of 1 mL/min. Samples of essential oils were desolved in pentane, and injection volume was 1µL. The split/splitless injector temperature was set at 280 °C. The investigated samples were injected in the split mode with a split ratio at 40.8:1. Electron-impact mass spectrometric data (EI-MS; 70 eV) were acquired in scan mode over the m/z range 35–550. The mass spectrometry ion source temperature was 230 °C, while the temperature of MS quadrupole was set at 150 °C. Solvent delay time of 3 min. After the separation, the components were identified based on the comparison of their relative retention index and compared with the library mass spectral database (Wiley and NIST databases). The percentage composition of compounds (relative quantity; amounts higher than 0.1%) was measured based on the peak area [33]. The retention indices were experimentally determined by injecton of standard n-alkanes (C_6_–C_34_) under the same chromatographic conditions.

### 4.3. Evaluation of EOs Antifungal Potential

#### 4.3.1. Fungal Strains and Culture Media

In the current study, three strains of genus *Penicillium* (*P. expansum, P. crustosum, P. citrinum*), isolated from berry samples of *Vitis vinifera* (growing in vineyards localized in Slovakia), were employed. Consequently, the microscopic filamentous fungi were classified using a reference-based MALDI-TOF MS Biotyper, and validated by comparison with the taxonomic identification using 16S ribosomal RNA (16S rRNA) gene sequences analysis.

To prepare fungal media, the strains were inoculated in Sabouraud Dextrose Agar (SDA; Oxoid, Basingstoke, UK) and incubated for 5 days at 25 °C. Subsequently, small aliquots of the fungi were transferred to test tubes, each containing 3 mL of distilled water. The inoculum concentration was standardized by comparison with the 0.5 McFarland scale (1.5 × 10^8^ CFU/mL).

#### 4.3.2. In Vitro Antifungal Activity of EOs

Evaluation of the in vitro antifungal activity of the EOs was performed using the agar disc diffusion method, according to Valková et al. [33] with minor modifications. For this purpose, an aliquot of 100 µL of culture media was inoculated on SDA. Then, the discs of filter paper (6 mm) were impregnated with 10 μL of each EO sample (in four concentrations: 62.5, 125, 250, and 500 μL/L), and applied on the SDA surfaces. Fungi were incubated aerobically at 25 ± 1 °C for 5 days. After the incubation, diameters of the inhibition zones in mm were measured. The values for inhibitory activity increased in the following manner: weak antifungal activity (5–10 mm) < moderate antifungal activity (10–15 mm) < very strong antifungal activity (zone > 15 mm).

#### 4.3.3. In Situ Antifungal Activity of EOs

All three fungal strains (*P. expansum*, *P. crustosum,* and *P. citrinum*) were used to evaluate the antifungal activity of the EOs in situ.

#### 4.3.4. Food Models

Three frequently consumed food products, i.e., bread, carrot, and potato were applied as substrates for the growth of the fungi. Among them, white bread was developed in the Laboratory of Cereal Technologies (Research Center AgroBioTech, SUA in Nitra) according to the methodology described in the study by Valková et al. [72]. The vegetables were purchased at the local market (Nitra, Slovakia).

#### 4.3.5. Moisture Content and Water Activity of Food Models

To predict the suitability of substrates for fungal growth, moisture content (MC) and water activity (aw) were determined, as reported by Valková et al. [73].

#### 4.3.6. Vapour Contact Assay

The experiment itself was performed as reported by Valková et al. [33]. After cooling, the bread slices with a thickness of 15 mm were transferred into glass jars (Bormioli Rocco, Fidenza, Italy; 500 mL). The inoculum of tested strains was applied by stabbing three times with an injection pin on the bread substrate. Then, a sterile filter paper disc (60 mm) was placed under the jar top, and 100 µL of the EOs in concentrations of 62.5, 125, 250, and 500 µL/L (diluted in ethyl acetate) were applied to it. The control bread was not treated with the EOs. Finally, the jars were hermetically closed and stored in an incubator for 14 days at 25 ± 1 °C. For vegetables (carrot and potato) used as food models, the methodology was slightly modified. Firstly, sliced carrot and potato (5 mm) were placed on the bottom of Petri dishes (PDs), and the inoculum was applied by stabbing one time with an injection pin on the vegetable surface. Further, 10 µL of the EOs (in the same four concentrations) was applied on the sterile filter paper disc (60 mm), then, it was placed at the top of PD. Subsequently, PDs were hermetically closed using parafilm and cultivated at 25 °C for 14 days.

#### 4.3.7. Determination of Fungal Growth Inhibition

In situ fungal growth was determined using stereological methods. In this concept, the volume density (Vv) of visible fungal colonies was firstly established using ImageJ software counting the points of the stereological grid hitting the colonies (P) and those (p) falling to the reference space (growth substrate used: bread, carrot, and potato). The volume density of strain colonies was consequently calculated as follows: Vv (%) = P/p. Finally, the antifungal potential of the EOs was expressed as the percentage of fungal growth inhibition (FGI) according to the formula FGI = [(C − T)/C] × 100, where C and T is the growth of fungal strains (expressed as Vv) in the control and treatment group, respectively [33].

### 4.4. Data Processing

The data were submitted to one-way analysis of variance (ANOVA) and the means were compared by the Tukey test at 5% of probability using statistical software Prism 8.0.1 (GraphPad Software, San Diego, CA, USA). All analyses were performed in triplicate.

## 5. Conclusions

The present research was carried out to analyze the presence of volatile profile and antifungal efficacies of commercial REO, NEO, and FEO against *Penicillium* strains isolated from berry fruits. Our findings revealed a variable chemical profile of analyzed EO samples with linalool (REO), 1,8-cineole (NEO), and α-pinene (FEO) as principal compounds of their composition. Regarding the in vitro antifungal activities, the EOs were effective in inhibiting the growth of all *Penicillium* strains (*P. expansum*, *P. citrinum*, and *P. crustosum*) in concentrations more than 125 µL/L, whereas the very strong inhibitory effect (16.00 ± 1.00 mm) was detected for the highest REO concentration against *P. citrinum*. Furthermore, the results from the estimation of MC and aw in the food substrates (bread, carrots, potatoes) showed a good growth potential of microscopic filamentous fungi. Similar trends, as found in in vitro antifungal analyses, were also observed in those performed on food models (in situ), indicating dose-dependent antifungal action of all EOs with the highest mycelial growth inhibition being recorded in their highest concentrations against all *Penicillium* spp. inoculated on all food models employed. In summary, our obtained data suggest that REO, NEO, and FEO have promising perspectives as innovative natural agents for application in the storage of food products (including bakery products and vegetables) to prolong their shelf life. To explain the utilization of our EOs as antifungal additives, we plan to perform a sensory evaluation of the food models investigated to reveal which effective concentrations of the EOs are still acceptable for the product consumers. Moreover, our findings complement our previous studies which contribute to create a more comprehensive overview of the biological properties of diverse, commercially available EOs purchased from the same company, Hanus Ltd. (Nitra, Slovakia).

## Figures and Tables

**Table 1 molecules-27-03101-t001:** Chemical composition of analyzed EOs.

No	Compound ^a^	REO (%)	NEO (%)	FEO (%)	RI	RI
					(lit.)	(calc.) ^b^
1	santene	/	/	1.6	888	889
2	2-bornene	/	/	0.2	907	909
3	tricyclene	/	/	3.1	926	924
4	α-thujene	Tr ^c^	tr	/	930	926
5	α-pinene	1	4.8	25.2	939	938
6	β-fenchene	/	/	tr	940	942
7	α-fenchene	/	/	2.6	952	947
8	camphene	0.1	0.5	13.4	954	948
9	benzaldehyde	/	tr	/	960	958
10	sabinene	tr	0.4	tr	975	977
11	β-pinene	0.9	2.2	18.3	979	980
12	β-myrcene	0.1	0.8	0.6	990	992
13	α-phellandrene	/	tr	tr	1002	1004
14	pseudolimonene	/	/	tr	1003	1003
15	δ-3-carene	/	tr	0.7	1011	1009
16	α-terpinene	tr	tr	0.1	1017	1016
17	*p*-cimene	/	3.2	/	1024	1023
18	*o*-cymene	1.3	/	1.5	1026	1026
19	α-limonene	1.3	6.4	18.1	1029	1028
20	β-phellandrene	/	/	1.1	1029	1030
21	1,8-cineole	16.9	40.8	/	1031	1033
22	(*E*)-β-ocimene	/	tr	/	1050	1047
23	γ-terpinene	1.5	1.8	tr	1059	1060
24	cis-linalool oxide	1.2	/	/	1072	1074
25	α-terpinolene	0.2	1.2	0.4	1088	1088
26	trans-linalool oxide	1.4	/	/	1086	1089
27	linalool	47.5	/	/	1096	1098
28	α-thujone	0.4	/	/	1102	1101
29	β-thujone	tr	/	/	1114	1114
30	cis-limonene oxide	/	/	0.2	1136	1136
31	trans-pinocarveol	/	/	0.2	1139	1140
32	trans-verbenol	/	/	tr	1144	1145
33	camphor	1.3	tr	/	1146	1148
34	menthone	/	tr	/	1152	1151
35	iso-menthone	/	tr	/	1162	1162
36	pinocarvone	/	/	tr	1164	1163
37	borneol	0.3	/	0.1	1169	1170
38	menthol	/	tr	/	1171	1173
39	4-terpinenol	2.9	1.5	tr	1171	1178
40	*p*-cymen-8-ol	/	/	tr	1182	1183
41	α-terpineol	5	14.6	0.6	1188	1189
42	verbenone	/	/	tr	1208	1208
43	endo-fenchyl acetate	/	/	0.2	1220	1221
44	nerol	0.2	/	/	1229	1227
45	carvone	/	/	tr	1243	1241
46	linalool acetate	0.4	/	/	1257	1255
47	2-phenyl ethyl acetate	tr	/	/	1258	1258
48	geranial	tr	/	/	1267	1263
49	(*E*)-cinnamaldehyde	tr	/	/	1270	1269
50	bornyl acetate	/	/	2.8	1285	1286
51	isobornyl acetate	tr	/	/	1285	1287
52	methyl geranate	0.1	/	/	1324	1321
53	α-terpinyl acetate	/	2.4	/	1349	1341
54	isoledene	0.4	/	/	1376	1371
55	α-copaene	/	tr	tr	1376	1379
57	longifolene	/	/	0.5	1407	1408
56	α-gurjunene	0.3	tr	/	1409	1408
58	(*E*)-caryophyllene	0.8	2.4	7	1419	1422
59	β-gurjunene	0.6	/	/	1433	1427
60	β-humulene	0.2	/	/	1438	1436
61	2-phenyl propyl isobutanoate	0.5	/	/	1440	1438
62	aromadendrene	5.3	0.6	/	1441	1443
63	α-humulene	/	tr	0.2	1454	1456
64	α-amorphene	/	tr	tr	1484	1485
65	β-selinene	/	tr	/	1490	1490
66	α-selinene	0.3	tr	/	1492	1492
67	valencene	0.4	/	/	1496	1497
68	ledene	1.5	1.6	/	1496	1498
69	bicyclogermacrene	0.8	/	/	1500	1503
70	α-muurolene	0.1	/	tr	1500	1504
71	δ-amorphene	0.1	/	/	1512	1513
72	δ-cadinene	0.5	0.5	/	1523	1525
73	cis-calamenene	0.1	/	/	1529	1530
74	α-cadinene	/	/	tr	1538	1542
75	palustrol	0.9	tr	/	1568	1570
76	spathulenol	0.5	/	/	1578	1577
77	caryophyllene oxide	1.2	tr	1	1583	1583
78	viridiflorol	0.3	12	/	1592	1593
79	widdrol	0.3	/	/	1599	1601
80	rosifoliol	0.2	/	/	1600	1604
81	ledol	/	1.7	/	1602	1605
82	α-cadinol	/	tr	/	1654	1656
	total	99.3	99.4	99.7		

^a^ Identified compounds; ^b^ values for retention indices on HP-5MS column; ^c^ tr—compounds identified in amounts less than 0.1%.

**Table 2 molecules-27-03101-t002:** Total amount of volatiles presented in percentage for each class of compounds.

Class of Compounds	REO (%)	NEO (%)	FEO (%)
non-terpenic compounds	0.5	tr	1.8
*hydrocarbons*	/	/	1.8
*aromatic compounds*	0.5	tr	/
monoterpenes	84	80.6	89.2
*monoterpene hydrocarbons*	6.4	21.3	85.1
*oxygenated monoterpenes*	77.6	59.3	4.1
monoterpene epoxide	19.5	40.8	0.2
monoterpene alcohols	55.9	16.1	0.9
monoterpene ketones	1.7	tr	tr
monoterpene esters	0.5	2.4	3
sesquiterpenes	14.8	18.8	8.7
*sesquiterpene hydrocarbons*	11.4	5.1	7.7
*oxygenated sesquiterpenes*	3.4	13.7	1
sesquiterpene alcohols	2.2	13.7	/
sesquiterpene epoxides	1.2	tr	1
total	99.3	99.4	99.7

**Table 3 molecules-27-03101-t003:** Antifungal activity of EO samples in analyzed concentrations (inhibition zone in mm).

	*P. expansum*				*P. citrinum*				*P. crustosum*			
Con. (µL/L)	62.5	125	250	500	62.5	125	250	500	62.5	125	250	500
REO	0.00 ± 0.00 ^aA^	0.00 ± 0.00 ^aA^	3.67 ± 0.58 ^aB^	10.33 ± 0.58 ^aC^	2.68 ± 0.58 ^aA^	5.33 ± 1.53 ^aB^	9.33 ± 0.58 ^aC^	16.00 ± 1.00 ^aD^	4.67 ± 0.58 ^aA^	5.33 ± 0.58 ^aA^	6.57 ± 0.58 ^aB^	10.67 ± 0.58 ^aC^
NEO	0.00 ± 0.00 ^aA^	0.00 ± 0.00 ^aA^	8.00 ± 1.00 ^bB^	11.00 ± 1.00 ^aC^	3.33 ± 0.58 ^aA^	4.33 ± 0.58 ^aAB^	5.33 ± 0.58 ^bB^	10.00 ± 1.00 ^bC^	0.00 ± 0.00 ^bA^	0.00 ± 0.00 ^bA^	4.67 ± 0.58 ^bB^	7.33 ± 0.58 ^bC^
FEO	0.00 ± 0.00 ^aA^	2.33 ± 0.58 ^bB^	3.33 ± 0.58 ^aB^	5.67 ± 1.15 ^bC^	4.00 ± 1.00 ^aA^	5.33 ± 0.58 ^aA^	7.00 ± 1.00 ^cB^	7.67 ± 0.58 ^cB^	0.00 ± 0.00 ^bA^	0.00 ± 0.00 ^bA^	5.67 ± 1.15 ^abB^	8.33 ± 0.58 ^bC^

Note: Mean ± standard deviation. REO: rosalina essential oil; NEO: niaouli essential oil; FEO: fir essential oil. Values in the same column with different small letters, and those in the same row (for the same type of fungi strains) with different upper-case letters, are significantly different (*p* < 0.05). Con.—concentration; 0.00—total growth; N—without growth.

**Table 4 molecules-27-03101-t004:** Moisture content and water activity in food models analyzed.

Parameters	Bread	Carrot	Potato
MC (%)	43.12 ± 0.35 ^a^	86.83 ± 0.42 ^b^	81.55 ± 1.65 ^c^
a_w_	0.942 ± 0.001 ^a^	0.945 ± 0.002 ^b^	0.946 ± 0.002 ^c^

Note: Mean ± standard deviation. MC—moisture content; a_w_—water activity. Values in the same row with different small letters are significantly different (*p* < 0.05).

**Table 5 molecules-27-03101-t005:** In situ antifungal activity of the EO samples in analyzed concentrations against the growth of selected *Penicillium* spp. inoculated on bread.

Fungi Strain	*P. expansum*				*P. citrinum*				*P. crustosum*			
Con. (µL/L)	62.5	125	250	500	62.5	125	250	500	62.5	125	250	500
REO	8.14 ± 0.95 ^aA^	1.20 ± 0.80 ^aB^	29.27 ± 1.56 ^aC^	98.50 ± 3.82 ^aD^	−2.81 ± 1.40 ^aA^	24.36 ± 3.57 ^aB^	95.31 ± 4.03 ^aC^	89.50 ± 3.73 ^aC^	−30.43 ± 4.11 ^aA^	−47.37 ± 5.98 ^aB^	90.02 ± 3.42 ^aC^	97.32 ± 4.51 ^aC^
NEO	18.21 ± 1.65 ^bA^	5.11 ± 2.33 ^bB^	13.07 ± 3.92 ^bA^	46.53 ± 4.85 ^bC^	4.71 ± 1.19 ^bA^	26.55 ± 1.37^aB^	23.89 ± 2.64 ^bB^	79.84 ± 3.66 ^bC^	13.87 ± 3.79 ^bA^	11.90 ± 1.61 ^bA^	84.04 ±6.97^aB^	95.60 ± 4.12 ^aC^
FEO	19.92 ± 6.51 ^bA^	19.22 ± 6.76 ^cA^	18.62 ± 4.87 ^bA^	81.72 ± 4.17 ^cB^	50.87 ± 1.47 ^cA^	49.95 ± 1.73 ^bA^	41.23 ± 1.18 ^cB^	87.94 ± 10.3 ^abC^	54.25 ± 4.61 ^cA^	25.64 ± 1.15 ^cB^	37.62 ± 1.65 ^bC^	45.13 ± 1.81 ^bD^

Note: Mean ± standard deviation. REO: rosalina essential oil; NEO: niaouli essential oil; FEO: fir essential oil. Values in the same column with different small letters, and those in the same row (for the same type of fungi strains) with different upper-case letters, are significantly different (*p* < 0.05). Con.—concentration.

**Table 6 molecules-27-03101-t006:** In situ antifungal activity of the EO samples in analyzed concentrations against the growth of selected *Penicillium* spp. inoculated on carrot.

Fungi Strain	*P. expansum*				*P. citrinum*				*P. crustosum*			
Con. (µL/L)	62.5	125	250	500	62.5	125	250	500	62.5	125	250	500
REO	52.17 ± 1.27 ^aA^	60.00 ± 1.55 ^aB^	100.00 ± 0.00 ^aC^	100.00 ± 0.00 ^aC^	54.05 ± 1.35 ^aA^	72.73 ± 6.10 ^aB^	87.84 ± 7.71 ^aC^	100.00 ± 0.00 ^aD^	54.88 ± 2.24 ^aA^	59.74 ± 3.47 ^aA^	96.51 ± 4.41 ^aB^	100.00 ± 0.00 ^aB^
NEO	40.35 ± 7.56 ^bA^	0.00 ± 0.00 ^bB^	39.39 ± 8.89 ^bA^	72.22 ± 5.4 5^bC^	14.75 ± 2.94 ^bA^	26.32 ± 7.57 ^bB^	6.90 ± 1.11 ^bC^	62.00 ± 1.98 ^bD^	0.00 ± 0.00 ^bA^	0.00 ± 0.00 ^bA^	2.78 ± 0.48 ^bB^	14.87 ± 1.65 ^bC^
FEO	48.44 ± 6.36 ^abA^	8.64 ± 3.69 ^cB^	9.86 ± 5.82 ^cB^	97.67 ± 6.22 ^aC^	7.69 ± 2.76 ^cA^	0.00 ± 0.00 ^cB^	9.80 ± 3.58 ^bA^	41.00 ± 4.97 ^cC^	0.00 ± 0.00 ^bA^	20.00 ± 8.94 ^cB^	39.68 ± 4.19 ^cC^	64.47 ± 6.37 ^cD^

Note: Mean ± standard deviation. REO: rosalina essential oil; NEO: niaouli essential oil; FEO: fir essential oil. Values in the same column with different small letters, and those in the same row (for the same type of fungi strains) with different upper-case letters, are significantly different (*p* < 0.05). Con.—concentration.

**Table 7 molecules-27-03101-t007:** In situ antifungal activity of the EO samples in analyzed concentrations against the growth of selected *Penicillium* spp. inoculated on potato.

Fungi Strain	*P. expansum*				*P. citrinum*				*P. crustosum*			
Con. (µL/L)	62.5	125	250	500	62.5	125	250	500	62.5	125	250	500
REO	23.21 ± 5.41 ^aA^	4.00 ± 1.17 ^aB^	53.18 ± 9.83 ^aC^	83.02 ± 2.74 ^aD^	0.00 ± 0.00 ^aA^	0.00 ± 0.00 ^aA^	86.32 ± 1.83 ^aB^	76.67 ± 6.44 ^aC^	0.00 ± 0.00 ^aA^	29.17 ± 2.86 ^aB^	40.82 ± 4.89 ^aC^	70.97 ± 6.37 ^aD^
NEO	0.00 ± 0.00 ^bA^	0.00 ± 0.00 ^bA^	0.00 ± 0.00 ^bA^	71.43 ± 9.59 ^aB^	0.00 ± 0.00 ^aA^	0.00 ± 0.00 ^aA^	0.00 ± 0.00 ^bA^	86.67 ± 8.96 ^aB^	0.00 ± 0.00 ^aA^	0.00 ± 0.00 ^bA^	15.79 ± 1.32 ^bB^	92.00 ± 6.83 ^bC^
FEO	7.81 ± 1.47 ^cA^	9.62 ± 1.45 ^cA^	82.76 ± 3.12 ^cB^	98.31 ± 8.92 ^bC^	10.91 ± 1.36 ^bA^	1.79± 0.62 ^bB^	7.32 ± 1.33 ^cC^	18.75 ± 2.26 ^bD^	0.00 ± 0.00 ^aA^	0.00 ± 0.00 ^bA^	2.00 ± 1.17 ^cB^	9.59 ± 1.52 ^cC^

Note: Mean ± standard deviation. REO: rosalina essential oil; NEO: niaouli essential oil; FEO: fir essential oil. Values in the same column with different small letters, and those in the same row (for the same type of fungi strains) with different upper-case letters, are significantly different (*p* < 0.05). Con.—concentration.

## Data Availability

Not applicable.

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
