# Peer review of "Assessment of Chemical Composition and Anti-Penicillium Activity of Vapours of Essential Oils from Abies Alba and Two Melaleuca Species in Food Model Systems"

_molecules, 2022, doi:10.3390/molecules27103101_

Round 1

Reviewer 1 Report

  1. I suggest using Latin scientific names of plants in the abstract part instead of common names.
  1. The authors have to write about why they chose these EOs before the aims of your study in the abstract.
  1. Correct the names of EOs in the table to stract with capital letters.
  1. Correct Tables 1-3 to contain RT, RI calculated, and RI literature for each compound in these three tables.
  1. According to the major EOs components, I suggest adding a clear mechanism of action and molecular docking also a structure-activity relationship.
  1. The manuscript needs major grammar, typo, and editing corrections.

Author Response

Reviewer #1

Point 1: I suggest using Latin scientific names of plants in the abstract part instead of common names.

Response: Revised directly in the manuscript.

Point 2: The authors have to write about why they chose these EOs before the aims of your study in the abstract.

Response:  Revised directly in the manuscript.

Point 3: Correct the names of EOs in the table to stract with capital letters.

Response: Revised directly in the manuscript.

Point 4: Correct Tables 1-3 to contain RT, RI calculated, and RI literature for each compound in these three tables.

Response:  In accordance to the reviewer's suggestion in Table 1 that is now sum of Tables 1-3, we included RI calculated and literature. RT of compounds is missing due to column variability over time.

Point 5: According to the major EOs components, I suggest adding a clear mechanism of action and molecular docking also a structure-activity relationship.

Response: Added directly in the manuscript.

Point 6: The manuscript needs major grammar, typo, and editing corrections.

Response: Revised directly in the manuscript.

Reviewer 2 Report

Overall, this manuscript is well designed and well written with moderate English changes required, however results section needs to be improved regarding data presentation and interpretation. Below, are the specific comments.

Title

Too long, authors should change this title, in my opinion “Assessment of Potential Use of Essential Oils from Myrtaceae and Pinaceae Trees as Food Preservatives “, should be enough.

Abstract

Lines 17-19: “Antifungal activities of three commercial essential oils (EOs; 62.5 μL/L, 125 μL/L, 250 μL/L, and 500 μL/L): rosalina (REO), niaouli (NEO), and fir (FEO) in respect to their chemical profiles were evaluated in the present research. Our findings revealed….”

Antifungal activities of three commercial essential oils (EOs), rosalina (REO), niaouli (NEO), and fir (FEO), were evaluated in the present research in respect to their chemical profiles, over four different concentrations (62.5 μL/L, 125 μL/L, 250 μL/L, and 500 μL/L). The findings revealed….

Lines 20-24:From the in vitro antifungal determination it can be obvious that inhibition zones of Penicillium spp. mycelial growth varied from 00.00 ± 00.00 mm (no inhibitory effectiveness) to 16.00 ± 1.00 mm, indicating a very strong antifungal activity which was detected against P. citrinum after the highest REO concentration exposure.”

In vitro antifungal determinations showed that inhibition zones of Penicillium spp. mycelial growth ranges from no inhibitory effectiveness (00.00 ± 00.00 mm) to 16.00 ± 1.00 mm, indicating a very strong antifungal activity which was detected against P. citrinum after the highest REO concentration exposure

Lines 27-29: Our results allow for the prediction of the investigated EOs as promising innovative agents in the storage of different types of food products (bread, carrot and potato) in order to extend their shelf-life.”

These results indicate that the investigated EOs may be promising innovative agents in order to extend the shelf-life of different types of food products such as bread, carrot and potato.

Keywords

According to the brief title, I suggest: volatile compounds; in vitro antifungal activities; in situ efficacy; food model systems; disc diffusion method

Advise: authors should use keywords which are not in the title

Introduction

Line 41: “ … more and more resistant…”

increasingly resistant

Line 44: “…antifungal agents coatings, …”

antifungal agent coatings

Line 48: “Essential oils are uncolored fluids, principally containing the volatiles and aromatic substances ….”

Essential oils are uncolored fluids, composed of mainly volatiles and aromatic substances

Line 50: “Owing to a complex chemical profile, their application appears to be a viable way in prevention and elimination of antifungal food spoilage…”

Owing to the complex chemical profile….

Line 53-55: “Of these constituents, the most important are terpene hydrocarbons or terpenoid compounds and terpenoids designated as isoprenoids and their derivatives…”

Of these constituents, the most relevant for antifungal activities are terpenoid compounds and their derivatives, also designated as isoprenoids since classification of terpenoids is based on the number of isoprene units…

Line 55:For the extraction of EOs exist a variety of methods, exhibiting certain advantages and determining physicochemical and biological properties….”

There are several methods for the extraction of EOs, each exhibiting certain advantages and determining physicochemical and biological properties….

Line 72: “… also indicate its significant phytomedicine potency.”

also indicate its significant phytomedicine potential

Lines 73-78: “In this context, as well as in the continuation of our previous experiments to find new interesting naturally food antifungal agents, the major objective of the present study was to characterize three commercially available EOs, and to evaluate their chemical composition and antifungal efficacies against selected microscopic filamentous fungi of genus Penicillium (P.; P. expansum, P. citrinum, and P. crustosum). Ultimately, the obtained findings will predict their use in the food industry to extend the shelf-life of food products.”

In this context, as well as following our previous experiments to find new interesting naturally food antifungal agents, the aim of the present study is to characterize three commercially available EOs, by evaluating their chemical composition and antifungal efficacies against selected microscopic filamentous fungi of genus Penicillium (P.; P. expansum, P. citrinum, and P. crustosum). Finally, their use by the food industry to extend the shelf-life of food products will be evaluated on food based models (bread, carrot and potato).

Results

Tables 1, 2 and 3 should be converted in only one table (Table 1), it will be much easy to compare the EOs.

Section 2.1

It would be interesting if authors refer to how many compounds were identified in each EOs, from what group, and for that Table 1 (which includes table 1, 2 and 3) should be organized by terpene groups (monoterpenes alcohols, cetones…sesquiterpenes,…).

Section 2.2

Table 4 should be converted to 3 graphs, Figure 1a for REO, Figure 1b for NEO and Figure 1c for FEO.

Line 104: “… potencies ...” change for “ … activities …”

Line 110: “… in a following manner …” change for “… as follows: …”

Section 2.3

Authors need to choose if is model foods or food models, please choose and change throughout the manuscript

This section should have a brief introduction/statement for the purpose of these analyses.

Section 2.4

Figures 1, 2 and 3 should be deleted, it’s impossible to take any information from these figures. The information authors want to achieve with these figures is in Table 6, 7 and 8 which in turn should be converted preferably to 3 graphs similar as previous, Figure 2a for REO, Figure 2b for NEO and Figure 2c for FEO or in 9 graphs, which then originates also Figure 3 and 4. Please, do not forget the control.

Line 147: “ … decreased in the following manner: REO > FEO > NEO.” Change for “ … decreased in the following order: REO > FEO > NEO.”

Line 163: “ Fir EO…” should be FEO or there are any reason for calling Fir EO?

Discussion

Line 230: Fir EO again, why? It should not be FEO?

Lines 239-241: I can’t see the relevance of this paragraph to this work

Material and methods

Section 4.2

Information regarding column, temperature program, detector and injector temperatures, and others are missing.

Author Response

Reviewer #2

Overall, this manuscript is well designed and well written with moderate English changes required, however results section needs to be improved regarding data presentation and interpretation. Below, are the specific comments.

Response: Thank you very much for the favorable comment.

Title

Too long, authors should change this title, in my opinion “Assessment of Potential Use of Essential Oils from Myrtaceae and Pinaceae Trees as Food Preservatives “, should be enough.

Response: Revised directly in the manuscript.

Abstract

Lines 17-19: “Antifungal activities of three commercial essential oils (EOs; 62.5 μL/L, 125 μL/L, 250 μL/L, and 500 μL/L): rosalina (REO), niaouli (NEO), and fir (FEO) in respect to their chemical profiles were evaluated in the present research. Our findings revealed….”

Antifungal activities of three commercial essential oils (EOs), rosalina (REO), niaouli (NEO), and fir (FEO), were evaluated in the present research in respect to their chemical profiles, over four different concentrations (62.5 μL/L, 125 μL/L, 250 μL/L, and 500 μL/L). The findings revealed….

Response: Revised directly in the manuscript.

Lines 20-24: “From the in vitro antifungal determination it can be obvious that inhibition zones of Penicillium spp. mycelial growth varied from 00.00 ± 00.00 mm (no inhibitory effectiveness) to 16.00 ± 1.00 mm, indicating a very strong antifungal activity which was detected against P. citrinum after the highest REO concentration exposure.”

In vitro antifungal determinations showed that inhibition zones of Penicillium spp. mycelial growth ranges from no inhibitory effectiveness (00.00 ± 00.00 mm) to 16.00 ± 1.00 mm, indicating a very strong antifungal activity which was detected against P. citrinum after the highest REO concentration exposure

Response: Revised directly in the manuscript.

Lines 27-29“Our results allow for the prediction of the investigated EOs as promising innovative agents in the storage of different types of food products (bread, carrot and potato) in order to extend their shelf-life.”

These results indicate that the investigated EOs may be promising innovative agents in order to extend the shelf-life of different types of food products such as bread, carrot and potato.

Response: Revised directly in the manuscript.

Keywords

According to the brief title, I suggest: volatile compounds; in vitro antifungal activities; in situ efficacy; food model systems; disc diffusion method

Advise: authors should use keywords which are not in the title

Response: Revised directly in the manuscript. Thank you for your advice, we really appreciate it.

Introduction

Line 41: “ … more and more resistant…”

increasingly resistant

Response: Revised directly in the manuscript.

Line 44: “…antifungal agents coatings, …”

antifungal agent coatings

Response: Revised directly in the manuscript.

Line 48: “Essential oils are uncolored fluids, principally containing the volatiles and aromatic substances ….”

Essential oils are uncolored fluids, composed of mainly volatiles and aromatic substances

Response: Revised directly in the manuscript.

Line 50: “Owing to a complex chemical profile, their application appears to be a viable way in prevention and elimination of antifungal food spoilage…”

Owing to the complex chemical profile….

Response: Revised directly in the manuscript.

Line 53-55: “Of these constituents, the most important are terpene hydrocarbons or terpenoid compounds and terpenoids designated as isoprenoids and their derivatives…”

Of these constituents, the most relevant for antifungal activities are terpenoid compounds and their derivatives, also designated as isoprenoids since classification of terpenoids is based on the number of isoprene units…

Response: Revised directly in the manuscript.

Line 55: “For the extraction of EOs exist a variety of methods, exhibiting certain advantages and determining physicochemical and biological properties….”

There are several methods for the extraction of EOs, each exhibiting certain advantages and determining physicochemical and biological properties….

Response: Revised directly in the manuscript.

Line 72: “… also indicate its significant phytomedicine potency.”

also indicate its significant phytomedicine potential

Response: Revised directly in the manuscript.

Lines 73-78: “In this context, as well as in the continuation of our previous experiments to find new interesting naturally food antifungal agents, the major objective of the present study was to characterize three commercially available EOs, and to evaluate their chemical composition and antifungal efficacies against selected microscopic filamentous fungi of genus Penicillium (P.; P. expansum, P. citrinum, and P. crustosum). Ultimately, the obtained findings will predict their use in the food industry to extend the shelf-life of food products.”

In this context, as well as following our previous experiments to find new interesting naturally food antifungal agents, the aim of the present study is to characterize three commercially available EOs, by evaluating their chemical composition and antifungal efficacies against selected microscopic filamentous fungi of genus Penicillium (P.; P. expansum, P. citrinum, and P. crustosum). Finally, their use by the food industry to extend the shelf-life of food products will be evaluated on food based models (bread, carrot and potato).

Response: Revised directly in the manuscript.

Results

Tables 1, 2 and 3 should be converted in only one table (Table 1), it will be much easy to compare the EOs.

Response: Revised directly in the manuscript.

Section 2.1

It would be interesting if authors refer to how many compounds were identified in each EOs, from what group, and for that Table 1 (which includes table 1, 2 and 3) should be organized by terpene groups (monoterpenes alcohols, cetones…sesquiterpenes,…).

Response: In accordance to the reviewer's suggestion, we modified Table 1, 2 and 3 to create one table (Table 1) and added Table 2. that includes content of class of compounds)

Section 2.2

Table 4 should be converted to 3 graphs, Figure 1a for REO, Figure 1b for NEO and Figure 1c for FEO.

Response: Despite the reviewer's suggestion, we consider demonstration of the obtained data in the form of the table to be more clarified. Following our previous experience and experimental works considering this issue (Galovičová et al., 2021a; Galovičová et al., 2021b; Kačániová et al., 2021; Valková et al., 2022), we have chosen to present the results in the form of the tables. Anyway, we thanks for an interesting alternative, we will certainly take it into consideration in our future work.

References:

Galovičová, L., Borotová, P., Valková, V., Vukovic, N. L., Vukic, M., Terentjeva, M., ... & Kačániová, M. (2021a). Thymus serpyllum Essential Oil and Its Biological Activity as a Modern Food Preserver. Plants, 10(7), 1416.

Galovičová, L., Borotová, P., Valková, V., Vukovic, N. L., Vukic, M., Štefániková, J., ... & Kačániová, M. (2021b). Thymus vulgaris essential oil and its biological activity. Plants, 10(9), 1959.

Kačániová, M., Galovičová, L., Valková, V., Tvrdá, E., Terentjeva, M., Žiarovská, J., ... & Kowalczewski, P. Ł. (2021). Antimicrobial and antioxidant activities of Cinnamomum cassia essential oil and its application in food preservation. Open Chemistry, 19(1), 214-227.

Valková, V., Ďúranová, H., Galovičová, L., Borotová, P., Vukovic, N. L., Vukic, M., & Kačániová, M. (2022). Cymbopogon citratus Essential Oil: Its Application as an Antimicrobial Agent in Food Preservation. Agronomy, 12(1), 155.

Line 104: “… potencies ...” change for “ … activities …”

Response: Revised directly in the manuscript.

Line 110: “… in a following manner …” change for “… as follows: …”

Response: Revised directly in the manuscript.

Section 2.3

Authors need to choose if is model foods or food models, please choose and change throughout the manuscript

Response: Revised directly throughout the manuscript.

This section should have a brief introduction/statement for the purpose of these analyses.

Response: Revised directly in the manuscript.

Section 2.4

Figures 1, 2 and 3 should be deleted, it’s impossible to take any information from these figures. The information authors want to achieve with these figures is in Table 6, 7 and 8 which in turn should be converted preferably to 3 graphs similar as previous, Figure 2a for REO, Figure 2b for NEO and Figure 2c for FEO or in 9 graphs, which then originates also Figure 3 and 4. Please, do not forget the control.

Response: Despite the reviewer's suggestion, we consider demonstration of the obtained data in the form of the table to be more clarified. Following our previous experience and experimental works considering this issue (Valková et al., 2021; Galovičová et al., 2021; Kačániová et al., 2021; Valková et al., 2022; Galovičová et al., 2022), we have chosen to present the results in the form of the tables. Anyway, we thanks for an interesting alternative, we will certainly take it into consideration in our future work.

References:

Galovičová, L., Borotová, P., Valková, V., Vukovic, N. L., Vukic, M., Terentjeva, M., ... & Kačániová, M. (2021a). Thymus serpyllum Essential Oil and Its Biological Activity as a Modern Food Preserver. Plants, 10(7), 1416.

Galovičová, L., Borotová, P., Valková, V., Vukovic, N. L., Vukic, M., Štefániková, J., ... & Kačániová, M. (2021b). Thymus vulgaris essential oil and its biological activity. Plants, 10(9), 1959.

Kačániová, M., Galovičová, L., Valková, V., Tvrdá, E., Terentjeva, M., Žiarovská, J., ... & Kowalczewski, P. Ł. (2021). Antimicrobial and antioxidant activities of Cinnamomum cassia essential oil and its application in food preservation. Open Chemistry, 19(1), 214-227.

Valková, V., Ďúranová, H., Galovičová, L., Borotová, P., Vukovic, N. L., Vukic, M., & Kačániová, M. (2022). Cymbopogon citratus Essential Oil: Its Application as an Antimicrobial Agent in Food Preservation. Agronomy, 12(1), 155.

Line 147: “ … decreased in the following manner: REO > FEO > NEO.” Change for “ … decreased in the following order: REO > FEO > NEO.”

Response: Revised directly throughout the manuscript.

Line 163: “ Fir EO…” should be FEO or there are any reason for calling Fir EO?

Response: Revised directly throughout the manuscript. The reason was only not to start the sentence with the abbreviation.

Discussion

Line 230: Fir EO again, why? It should not be FEO?

Response: Revised directly throughout the manuscript.

Lines 239-241: I can’t see the relevance of this paragraph to this work

Response: The paragraph was removed from the manuscript.

Material and methods

Section 4.2

Information regarding column, temperature program, detector and injector temperatures, and others are missing.

Response: According to the reviewer's suggestion in the section 4.2. (Lines 361-370 and 373-375) we added missing data.

Round 2

Reviewer 1 Report

The authors conducted all the required corrections

Author Response

thanks